# The Prediction of Survival after Surgical Management of Bone Metastases of the Extremities—A Comparison of Prognostic Models

**Ofir Ben Gal [1,\*], Terrence Chi Fang Soh [2], Sarah Vaughan [1], Viswanath Jayasanker [1], Ashish Mahendra [1] and Sanjay Gupta [1]**

[1] Department of Musculoskeletal Oncology, Glasgow Royal Infirmary, 84 Castle Street, Glasgow G4 0SF, UK; sarah.vaughan@ggc.scot.nhs.uk (S.V.); viswanath.jayasanker@ggc.scot.nhs.uk (V.J.); ashish.mahendra@ggc.scot.nhs.uk (A.M.); sanjay.gupta@ggc.scot.nhs.uk (S.G.)

[2] University Hospital Hairmyres, 218 Eaglesham Rd, East Kilbride, Glasgow G75 8RG, UK; fangt@nhsl.lanarkshir.scot.nhs.uk

[\*] Correspondence: ofir.bengal@ggc.scot.nhs.uk; Tel.: +44-07535819383

**Abstract:** Individualized survival prognostic models for symptomatic patients with appendicular metastatic bone disease are key to guiding clinical decision-making for the orthopedic surgeon. Several prognostic models have been developed in recent years; however, most orthopedic surgeons have not incorporated these models into routine practice. This is possibly due to uncertainty concerning their accuracy and the lack of comparison publications and recommendations. Our aim was to conduct a review and quality assessment of these models. A computerized literature search in MEDLINE, EMBASE and PubMed up to February 2022 was done, using keywords: "Bone metastasis", "survival", "extremity" and "prognosis". We evaluated each model's performance, assessing the estimated discriminative power and calibration accuracy for the analyzed patients. We included 11 studies out of the 1779 citations initially retrieved. The 11 studies included seven different models for estimating survival. Among externally validated survival prediction scores, PATHFx 3.0, 2013-SPRING and potentially Optimodel were found to be the best models in terms of performance. Currently, it is still a challenge to recommend any of the models as the standard for predicting survival for these patients. However, some models show better performance status and other quality characteristics. We recommend future, large, multicenter, prospective studies to compare between PATHfx 3.0, SPRING 2013 and OptiModel using the same external validation dataset.

**Keywords:** bone metastases; survival; extremity; prognosis; long bone metastases; prognostic score; surgery





## 1. Introduction

Accurately predicting the life expectancy of patients with metastatic bone disease of the extremity is essential as therapeutic strategies largely depend on it [1]. In this setting, a multidisciplinary team treats the patient with the intention of improving function, reducing pain, minimizing the number of procedures and rehabilitation time, hence allowing the maximum quality of life for the remaining period of survival [1,2]. Generally, patients with short term life expectancy will most likely be treated with supportive care, short-term radiotherapy and either no surgery or internal fixation surgery as its related complications typically occur more than a year after treatment. Patients with long-term life expectancy would potentially benefit from chemotherapy, radiotherapy and prosthetic reconstruction, which is associated with a decreased mechanical failure rate after 1 year [1,3–5].

An exact estimation of survival, however, is difficult and physicians tend to be inaccurate up to a factor of three according to Chen et al. or only in 18% of cases according to Nathan et al. [6–8]. This has prompted the development of more objective means of

estimating life expectancy in this potentially terminally ill patient population. Evidence has shown that surgical management of MBD achieves both pain relief and maintains function in almost 90% of patients [9]. To accomplish this, an orthopedic surgeon must base their treatment decisions on various factors, such as the location and physiology of the lesion, the desired mechanical properties and limitations of the implant, the patient's wishes and their prognostic survival time.

An expected survival of less than 4–6 weeks is a relatively well-agreed contraindication for any surgical management. Most authors consider an expected survival of 3–12 months as an appropriate threshold for the consideration of less invasive surgical reconstruction procedures that do not need prolonged rehabilitation and 12 months or more as the "long" survivors, in which more invasive resection and reconstruction are indicated despite prolonged rehabilitation, taking into account the risk of local recurrence [1,10]. However, some consider six months of postoperative survival an indication for using more durable implants such as endoprostheses (EPR) rather than internal fixation (INF). EPR will more likely outlive the patient, minimizing the risk of implant breakage or non-union, specifically for lesions involving the peritrochanteric or subtrochanteric femur secondary to disease progression [11].

Various prognostic studies on specific primary cancers have been conducted but having different prediction score for each group of patients does not provide the generalized, easy-to-use approach essential for the orthopedic surgeon in this clinical scenario. In addition, tumor-specific prognostic scores do not provide large amounts of patients as more generalized studies do. However, the relative risk of each specific cancer can be an acceptable surrogate and must be taken under consideration [12–14].

Some landmark prognostic scoring systems are designed for MBD in the spine, alone or in the extremities as well, such as in Bauer and Wedin or Tokuhashi et al. [10,15]. However, our review, along with more recent studies, deal with them separately due to differences in morbidity and treatment [16,17].

Effective targeted and selected chemotherapeutic regimens, such as bone targeting agents, hormonal therapy, immunotherapy etc., in the treatment of advanced cancer were mainly available from 2005; thus, they might have a positive impact on survival and moreover, the development of targeted treatments for several primary tumors has subdivided primary tumors into different entities, which contributes to a relative progressive loss in the accuracy of prognostic scores that were generated earlier [18,19]. Chen et al. also stated that when creating estimation models for clinical outcomes, a small amount of modern data is more effective than a large amount of old data [20].

Therefore, several prognostic models have been developed in recent years to guide clinical decision-making for appropriate therapies in patients with appendicular metastasis. However, most orthopedic surgeons have not implemented these models in routine practice, possibly due to the high amount of uncertainty concerning their accuracy and applicability in clinical settings or the lack of comparison publications and recommendations. A rigorous assessment and summary are needed to clarify the overall quality and applicability of each model. Therefore, this review aims to update the existing evidence, conduct a critical appraisal and summarize the results of the individualized risk models which are used to estimate the survival of symptomatic MBD patients treated with surgery.

## 2. Materials and Methods

### 2.1. Data Source and Searches

The search included a computerized literature search on 1 February 2022 in MEDLINE (accessed through Ovid), EMBASE (accessed through Ovid) and PubMed. A systematic review of clinical studies about prognostic scores that estimate the survival of patients with appendicular bone metastases treated surgically was done using the keywords: "Bone metastasis", "survival", "extremity" and "prognosis". Further relevant publications were also checked in the reference lists of the selected papers. Papers other than the English language and publication dates before 2004 were excluded. Case studies

and reviews were also excluded (Figure 1). This research is based upon the assessment structure of the PRISMA statement and the Systematic Review registered in Open Science (10.17605/OSF.IO/UGC2T).

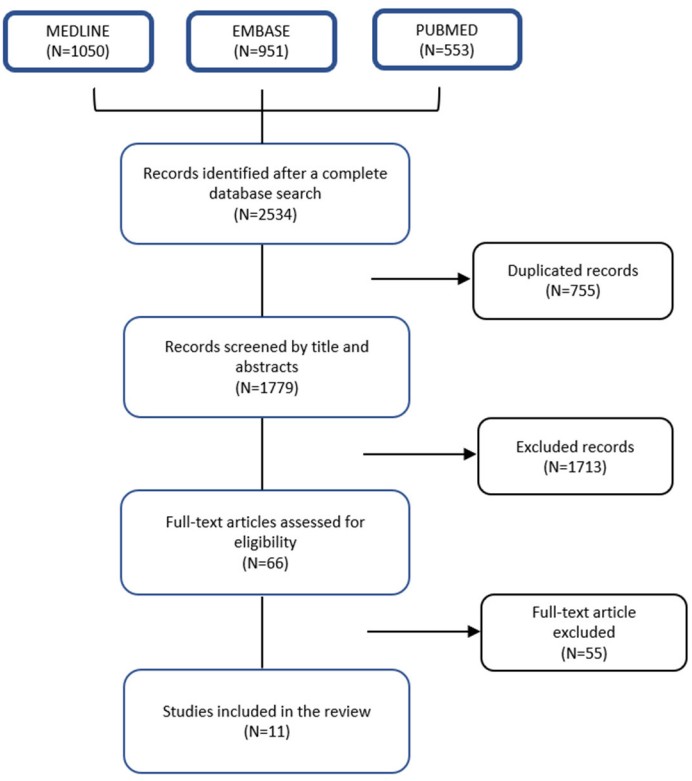

**Figure 1.** Study flowchart.

### 2.2. Data Selection

Articles were excluded if their investigation involved primary bone tumors, non-metastatic bone disease, axial bone metastasis and other treatment methods, such as ablation, radiotherapy, chemotherapy and bisphosphonates as their primary intervention. In addition, studies including databases only prior to 2005 were excluded. Publications that analyzed mixed populations treated with surgery and radiotherapy for symptomatic appendicular MBD were included.

Only original extensive papers of each risk model were included, excluding external validation studies or multiple publications that replicated previous models without any additional information such as a new design for collecting the inputs data, new risk factors or risk model method. Comparative external validation articles were included.

### 2.3. Data Synthesis and Analysis

We evaluated the model's validation by assessing overall performance, discriminative power and the calibration accuracy estimated for the analyzed patients. When available in the included publication, we extracted the area under the receiver operating characteristic curve (AUROC) and Brier score. Decision curve analysis was not included because it was poorly presented throughout the studies. The area under the curve ranged from 0.50 to 1.0, with 0.50 indicating complete coincidence and 1.0 indicating the highest discriminating capability. Calibration refers to an agreement between the predicted outcome and the actual outcome and perfect calibration has an intercept of 0 and a slope of 1 [8,21]. The Brier score indicates overall performance, with 0 as a perfect Brier score. We analyzed the performance status at the 3-, 6- and 12-month survival endpoint across the studies.

A narrative synthesis of each model has been conducted including a brief summary of key study characteristics. Methodological quality, accuracy and validation of individual

risk models are described in tables. Results are presented according to the original model that they reported.

Traditionally, scoring systems that estimate survival use a summary of each clinical or laboratory prognostic factor's relative value. Bauer and Katagiri [10,19] are classic scoring systems such as these.

Another way to estimate survival in patients with cancer is the nomogram, which is a unique figure that contains a set of lines, each line representing a prognostic factor whose value can be correlated with its weighted value on an assigned numerical line reading the nomogram and the sum of these points generates an individualized probability of survival. Machine learning is a subset of computer science and statistics utilizing automatically developing algorithms to recognize patterns in data and improve predictions [22]. It is capable of handling large amounts of data and different combinations of predictors to achieve a certain outcome.

## 3. Results

The database searches for primary studies retrieved 2534 citations. Duplicates were removed, leaving 1779 results of which, after a screening of the titles and abstracts, 66 were considered potentially relevant. These 66 studies were screened in full text. Eleven studies [16,17,23–31] met the inclusion criteria and were considered in the evidence synthesis and this is depicted in Figure 1.

The 11 studies include seven different models for estimating survival. Five studies discuss one model each and four studies discuss two models each that were an evolution of earlier versions. The final two studies are comparative publications. The models included in our analysis are as follows: Ratasvuori et al. (Scandinavian Sarcoma Group—referred to as the 7SSG score), Janssen et al. (referred to as Janssen score), Willeumier et al. (referred to as the OptiModel), Sorensen et al. (referred to as the SPRING 2008 and SPRING 2013 nomograms), Forsberg et al. (referred to as the PathFx 1.0 model, Anderson et al. (referred to as PATHfx 3.0 model), Thio et al. (referred to as SORG model) and Errani et al. (referred to as IOR score) [16,17,31] (Table 1).

All prognostic factors were produced by univariate [24,25] and multivariate Cox analysis [23,26,30,31]. The machine learning models used specific algorithms for this purpose. Four studies used traditional scoring systems such as survival estimation methods [23,26,30,31], two used nomograms [24,25,30], and a machine learning algorithm was the tool for three of the studies [27–30]. The most common methods to assess the performance of a prognostic score are considerations of the discrimination accuracy and the calibration score.

Thio et al. and Janssen et al. used a subset of the population that generated the data in order to assess external validation, while Ratasvuori et al., Willeumier et al., Sorensen et al. and Anderson et al. used an external population for that purpose. External validation was reported by subsequent comparative articles [16,17], using different populations including six models: OPTIModel, SPRING 2013, PATHfx 1.0, 7SSG, Janssen and IOR. SORG model was not externally validated. Only Errani et al. conducted a prospective study eliminating the bias due to the study design. Twelve-month survival was the only endpoint produced in all models.

### 3.1. Discrimination Accuracy and Calibration Score

All the studies that used AUC in their performance matrices showed reduced AUC in their externally validated cohort. AUCs and Brier scores for each validation set are shown in Table 2. These matrices for 3-, 6- and 12-month survivals have also been displayed in Figure 2A–C, respectively. All models have been shown to have mean AUCs ranging between 0.57 and 0.87 and mean Brier scores between 0.13 and 0.25.

**Table 1.** Summary of Included Scoring Models.

| Study ID—Ref | Type (y) | Patients (*n*) | Prognostic Factors (Number of Categories) | Intervention | Survival (m) | External Validation (*n*) |
|---|---|---|---|---|---|---|
| PATHfx. 1.0 Foresberg et al. (2011) [3] | Retrospective (1999–2009) | 815 | Age (y), Sex (2), Diagnostic group (3), Visceral metastases (2), Surgeon's estimate survival (4), Pathologic/Impending fx (2), Lymph node involvement (2), Skeletal metastases (2), Preoperative Hb (4) and lymphocyte count (4), ECOG performance score (3) | Surgery | 3,12 | Yes |
| PATHfx 3.0 Anderson et al. (2019) [29] | Retrospective (1999–2003) (2015–2018) | 397 | Age (y), Sex (2), Diagnostic group (3), Visceral metastases (2), Surgeon's estimate survival (4), Pathologic/Impending fx (2), Lymph node involvement (2), Skeletal metastases (2), Preoperative Hb (4) and lymphocyte count (4), ECOG performance score (3) | Surgery | 3,6,12,18,24 | Yes |
| SPRING 2008 Sorensen et al. (2016) [24] | Retrospective (2003–2008) | 121 | Diagnostic group (3), Hemoglobin (c), Visceral metastases (2), Bone metastases (2), Fracture/impending fracture (2), ASA group (2) and KPS (2) | EPR | 3,6,12 | No |
| SPRING 2013. Sorensen et al. (2018) [25] | Retrospective (2003–2013) | 270 | Diagnostic group (3), Hemoglobin (continuous), Visceral metastases (2), Bone metastases (2), fracture/impending fracture (2), ASA group (2) and KPS (2) | EPR | 3,6,12 | Yes |
| 7SSG. Ratasvuori et al. (2013) [23] | Retrospective (1999–2009) | 833 | Single bone metastases (2), Absence of visceral metastases (2), Primary tumor location in breast, kidney, thyroid, myeloma or lymphoma (2), KPS (2) | Surgery | 6,12 | Yes |
| OPTIModel. Willeumier et al. (2018) [26] | Retrospective (2000–2013) | 1520 | Diagnostic group (3), KPS (2), Visceral metastases (2) | RT or Surgery | 3,6,12 | Yes |
| SORG. Thio et al. (2019) [27] | Retrospective (1999–2017) | 1090 Training + Validation Datasets | Albumin (c), Neutrophil-to-lymphocyte ratio (c), Diagnostic group (3), ALP (c), Hemoglobin (c), Calcium (c) Absolute neutrophil count (c), WBC (C), Age, Platelet count (c), Visceral metastases (2), Sodium (c), Platelet-to-lymphocyte ratio | Surgery | 1,12 | No |
| Janssen et al. (2015) [30] | Retrospective (2009–2013) | 927 Training + Validation Datasets | Age 65 or older (2), Additional comorbidity (2), BMI less than 18.5 kg/m$^2$ (2), Tumor type other than breast, kidney, prostate, thyroid, myeloma or lymphoma (2), Bone metastases (2), Visceral metastases (2), Hemoglobin level 10 g/dL or less (2) | Surgery | 1,3,12 | Yes |
| IOR. Errani et al. (2021) [31] | Prospective (2015–2018) | 159 | Pathological C-reactive protein (2), Diagnostic group (2) | Surgery | 12 | Yes |

**Table 2.** Summary of performance metrics of externally validated models.

| Study Model | Study ID | Discrimination Accuracy (AUC) | | | Calibration (Brier Score) | | |
| --- | --- | --- | --- | --- | --- | --- | --- |
| | | 3 Months | 6 Months | 12 Months | 3 Months | 6 Months | 12 Months |
| PATHfx 1.0 | Alfaro et al. [17] | 0.62 (0.49, 0.73) | 0.66 (0.56, 0.75) | 0.54 (0.38, 0.68) | NA | NA | NA |
| | Meares et al. [16] | 0.70 (0.69, 0.7) | 0.70 (0.69, 0.70) | 0.71 (0.70, 0.71) | 0.23 (0.23, 0.23) | 0.23 (0.22, 0.23) | 0.19 (0.19, 0.19) |
| | Errani et al. [31] | | | 0.74 (NA) | | | |
| PATHfx 3.0 | Anderson et al. [29] (IBMR) | 0.77 (0.70, 0.84) | 0.77 (0.70, 0.83) | 0.78 (0.71, 0.85) | 0.20 (0.16, 0.23) | 0.20 (0.17, 0.24) | 0.18 (0.15, 0.22) |
| | Anderson et al. [29] (RT) | 0.83 (0.77, 0.90) | 0.79 (0.73, 0.86) | 0.79 (0.73, 0.86) | 0.14 (0.11, 0.17) | 0.20 (0.16, 0.24) | 0.20 (0.16, 0.24) |
| SPRING 2013 | Sorensen et al. [25] | 0.82 (0.73, 0.91) | 0.85 (0.76, 0.93) | 0.86 (0.77, 0.95) | 0.16 (0.12, 0.19) | 0.16 (0.13, 0.20) | 0.15 (0.12, 0.19) |
| | Meares et al. [16] | 0.66 (NA) | 0.68 (NA) | 0.76 (0.75, 0.76) | 0.25 (NA) | 0.26 (NA) | 0.19 (NA) |
| OPTIModel | Alfaro et al. [17] | 0.57 (0.44, 0.69) | 0.64 (0.54, 0.73) | 0.55 (0.39, 0.70) | NA | NA | NA |
| | Meares et al. [16] | 0.66 (NA) | 0.67 (NA) | 0.79 (0.78, 0.79) | 0.21 (NA) | 0.24 (NA) | 0.16 (0.16, 0.16) |
| | Errani et al. [31] | | | 0.751 (NA) | | | |
| IOR | Alfaro et al. [17] | NA | NA | 0.65 (0.50, 0.79) | NA | NA | NA |
| Janssen | Meares et al. [16] | 0.68 (NA) | NA | 0.71 (0.70, 0.71) | 0.21 (NA) | NA | 0.22 (0.22, 0.22) |
| 7SSG | Meares et al. [16] | 0.63 (NA) | 0.64 (NA) | 0.62 (NA) | 0.22 (NA) | 0.24 (NA) | 0.21 (NA) |
| | Errani et al. [31] | | | 0.72 (NA) | | | |

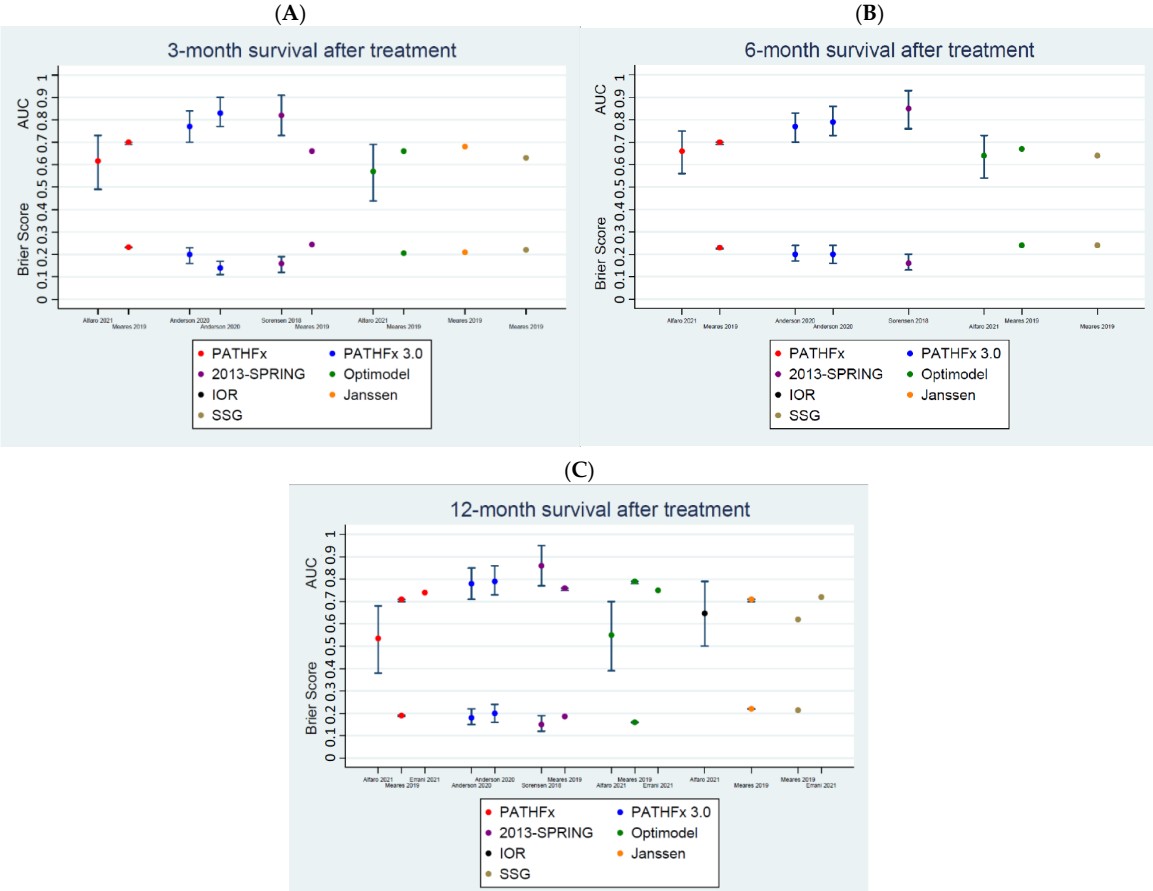

**Figure 2.** Performance status graphs, AUCs (area under the curve) and Brier scores. (**A**). Comparing metastatic bone disease predicting survival models for 3 months of survival across externally validated studies. (**B**). Comparing metastatic bone disease predicting survival models for 6 months of survival across externally validated studies. (**C**). Comparing metastatic bone disease predicting survival models for 12 months of survival across externally validated studies.

Firstly, 2013-SPRING and PATHFx 3.0 models have been shown to have the highest discrimination accuracies and best calibration scores (Figure 2A–C). This is true across all post-treatment survival time endpoints. However, the only exceptions are the 3- and 12-month post-treatment survival AUC [0.66 and 0.76 (0.75, 0.76)] and Brier scores [0.25 and 0.19] of 2013-SPRING, as validated by Meares et al., 2019 [16].

Secondly, the discrimination accuracies and calibration scores of the 2013-SPRING models seem to be better than that of PATHFx 3.0, at 6 and 12 months after treatment (Figure 2B,C). However, these matrices are comparable across the three models at the 3-month endpoint (Figure 2A), with the AUC [0.66] and Brier scores [0.25] of 2013-SPRING, as validated by Meares et al., 2019 [16], performing significantly worse.

Thirdly, AUCs [0.66 and 0.79 (0.78, 0.79)] and Brier scores [0.21 and 0.16 (0.36, 0.36)] of the Optimodel have been shown by Meares et al., 2019 [16] to be better than AUC [0.66 and 0.76 (0.75, 0.76)] and calibration scores [0.25 and 0.19] of the 2013-SPRING model at both the 3- and 12-month time periods (Figure 2A,C).

Lastly, the AUCs [0.87 (0.86, 0.88) and 0.85 (0.83, 0.86)] and calibration accuracies [0.13 (0.12, 0.14) and 0.16 (0.15, 0.16)] of SORG are significantly better than that of most other models at the 3- and 12-month endpoints (Table 2). However, it is important to note that these matrices reported by Thio et al., 2020 [27] are based on internally validated datasets.

### 3.2. Summary of the Models

### 3.2.1. SPRING 2013 Nomogram

In 2016, the SPRING 2008 [24] was published by Sorensen et al. [24] A survival prediction nomogram based solely on patients having bone resection and reconstruction surgeries due to MBD in the appendicular skeleton. The statistical analysis investigated data from 121 consecutive patients between 2003–2008 in Denmark. The authors used seven predictors of survival as variables: hemoglobin (mmol/L) (continuous variable), visceral metastases (yes/no), multiple bone metastases (yes/no), fracture status (impending fracture/fracture), Karnofsky score (<70% vs. ≥70%), ASA score (1 to 2/3 to 4) and primary cancer grouped according to Katagiri et al. [32] (slow growth/moderate growth/rapid growth), choosing the timing for the risk of death at 3, 6 and 12 months. One thousand crossmatch validations internally validated the model. In 2018, the same group successfully refitted the SPRING 2008, producing a free-to-use nomogram-type model predicting survival for 3, 6 and 12 months after surgery. The refitted SPRING 2013 used a modern, larger, consecutive cohort of patients (*n* = 270), the 'training cohort', treated from 2003–2013 with bone resection and reconstruction alone. However, it was externally validated in a prospective, Scandinavian, population-based cohort (*n* = 162), the 'validation cohort', who underwent both resection and reconstruction or internal fixation for metastatic bone disease in the extremities. The new model used the same prognostic factors as variables. A comparison of AUC ROC and Brier scores between the 2008-SPRING model and the 2013-SPRING model showed better performance of the refitted 2013 model when predicting survival, with good calibration in all three endpoints (3, 6, 12 months). AUC ROC curves of the external validation for survival predictions were 82% (95% CI, 73–91%) at 3 months, 85% (95% CI, 76–93%) at 6 months and 86% (95% CI, 77–95%) at 12 months [25]. In a study by Meares et al. [16] comparing prognostic models in 114 patients surgically treated for femoral MBD, SPRING 2013 nomograms were only sufficiently accurate at the 12-month period (AUC 0.76).

### 3.2.2. 7 SSG Score

In 2013, Ratasvuori et al. carried out a simple survival estimation score based on four prognostic variables in a series of 833 patients surgically treated for appendicular skeletal metastases between June 1999 and October 2009. The data were collected from the Scandinavian Sarcoma Group (SSG) Skeletal Metastasis Registry. The authors randomized 20% of patients (178) to a "testing set" and made the analysis with 651 patients, called a "training set". After sifting through non-significant variables such as age and pathologic fracture, only the following prognostic factors were found to have a positive effect on

survival: 1. single bone metastases, 2. absence of organ metastases, 3. primary tumor location in the breast, kidney, thyroid, myeloma or lymphoma and 4. Karnofsky score more than 70; each one of them received one point. The prognostic score is calculated, adding all the scores for individual factors and dividing the patients into five groups from one to four. According to the scoring system, the patients were allocated into three survival rate subgroups using the Kaplan Meier method: patients in group A survive quite reliably over 6 months and two-thirds of patients will survive over 12 months. Patients in group B are likely to survive over 3 months and half of the patients will survive over 6 months. Patients' survival even for 3 months is uncertain in group C. The testing set's survival rates were checked to validate the scoring system and they seem to correlate with survival rates in the training group [23]. The SSG scale was not sufficiently accurate in their application to the femoral MBD dataset according to Meares et al. [16].

### 3.2.3. Janssen Score, Nomogram, Model

In 2015, Janssen et al. compared the accuracy of a scoring system, a nomogram and boosting algorithms in predicting 30-, 90- and 365-day survival. A retrospective study based on 927 patients who underwent surgery for long bone metastases between 2009 and 2013. The authors found that older age, additional comorbidity (according to modified Charlson Comorbidity Index) [33], BMI less than 18.5 kg/m$^2$, primary tumor type with poor prognosis (dichotomizing the tumors into favorable or not based on the study by Katagiri et al.), multiple bone metastases, visceral metastases and lower hemoglobin level all independently decreased chances of survival and therefore, these were included in their models. The study identified high BMI, previously associated with survival in cancer patients [34], and comorbidity status as new prognostic factors but didn't use performance status (ECOG etc.) of the patient as one. The models were internally validated using five-fold cross-validation on multiple randomly selected training subsets (80%) of the database and externally validated on the remaining 20% of it. Performance was assessed using receiver operating characteristic (ROC) curves. The performance of the classic scoring system was found to have the lowest scores at all the prediction periods, while the nomogram had a moderate but stable accuracy when predicting survival at all periods. However, the boosting algorithm was better at predicting survival for all periods of the internal validation datasets, but it was of equal performance when applied to the external validation datasets. The author emphasizes that the nomogram is more useful for clinical practice [30]. Mearns et al. found the model sufficiently accurate at the 12-month period (AUC = 0.71) and lacked reliability at shorter time periods when estimating patient survival.

### 3.2.4. OptiModel

Developed at Leiden University, this classic scoring-type model uses data from a retrospective, multicenter study of 1520 patients with symptomatic long bone metastases treated with radiotherapy or surgery at six Dutch hospitals between 2000 and 2013. The model contains only three clinical independent variables: the clinical profile (favorable, moderate or unfavorable) based on the classification system established by Bollen et al. [35]; KPS is dichotomized into two groups and the presence of visceral and/or brain metastases. The study excluded primary hematological cancer and updated the subdivision in the clinical profile variable according to the new development of targeted treatments for several primary tumors. Combinations of the independent prognostic variables led to 12 prognostic categories and the median overall survival results of all categories were compared and visualized in a clinically applicable flowchart. Cut-off timeframes of 3, 6 and 12 months, relevant for decision-making in a clinical setting, were applied to narrow the 12 survival categories down to four clinically relevant categories. Median survival was A: 21.9 months (95% confidence interval [CI], 18.7 to 25.1 months), B: 10.5 months (95% CI, 7.9 to 13.1 months), C: 4.6 months (95% CI, 3.9 to 5.3 months) and D: 2.2 months (95% CI, 1.8 to 2.6 months). The applicability of the model was then externally validated by evaluating a series of 250 patients receiving surgical treatment between 2000 and 2013 at an Austrian

hospital and showed similar results between observed and expected survival. This model was incorporated into a freely accessible web application called OPTIModel, available since 2017 [26]. In 2019, Meares et al. demonstrated that OPTIModel was the most accurate model for predicting survival at 12-month and 24-month time periods.

### 3.2.5. SORG Model

Based on his earlier study regarding survival prediction of patients suffering from chondrosarcoma [36], in 2019, Thio et al. developed prediction models for 90-day and 1-year survival in patients who were surgically treated for bone metastases of the extremity using five different machine learning techniques. This retrospective study included 1090 patients between the years of 1999 and 2017. The data was divided into a training set (80%) and a validation set (20%). The training set was used to develop the models, while the validation set was used to internally validate the models. The authors first selected the variables for the algorithm by means of a 10-fold cross-validation of random forest algorithms in order to lower the variance and avoid overfitting of the model. The following factors were included as predictors associated with a 90-day likelihood of survival: albumin level, neutrophil-to-lymphocyte ratio, primary tumor group (as classified by Katagiri et al. [32]), alkaline phosphatase level, hemoglobin level, calcium level, absolute neutrophil count, white blood cell count, age and platelet count. Those factors for 1-year survival were albumin level, primary tumor type, hemoglobin level, neutrophil-to-lymphocyte ratio, alkaline phosphatase level, absolute lymphocyte count, presence of visceral metastases, sodium level, platelet-to-lymphocyte ratio and age. The predictive performance was assessed using 10-fold cross-validation of both training and validation sets. Performance was measured by discrimination (area under the curve), calibration (intercept and slope) and Brier score (overall performance). The five models showed good performance and no difference in discrimination and calibration in both sets. No external validation was made. The stochastic gradient-boosting algorithm was chosen for both 90-day and 1-year survival final prediction models and was made freely accessible through: HTTPs://sorg-apps.shinyapps.io/extremitymetssurvival/ (accessed on 3 April 2022). Global and individual explanations are provided there, in addition to the likelihood of survival for 90 days/one year and this helps the clinician understand which variables are associated with the provided predicted survival for a specific patient. The author instituted that in the chosen model, the most important factors associated with a greater risk of 90-day mortality were lower albumin level, higher neutrophil-to-lymphocyte ratio and rapid growth of the primary tumor, scaled from 0 to 100, the weighted importance of albumin, neutrophil-to-lymphocyte ratio and primary tumor category was 100, 75 and around 40, correspondingly. Whereas for 1-year mortality, there were lower albumin levels, rapid growth of primary tumors and lower hemoglobin levels with a relative importance of 100, 80 and 70, respectively [27].

### 3.2.6. PATHfx Model

Introduced in 2011 by Forsberg et al. and based on a Bayesian Belief Network capable of estimating 3- and 12-month survival for patients undergoing surgery for skeletal metastases, a training set was derived from 189 consecutive patients treated between 1999 and 2003 [3]. The model was initially comprised of ten prognostic features previously proven to affect the survival of these patients: age at the time of surgery (in the 12-month model), sex, indication for surgery (impending or completed pathologic fracture), number of bone metastases (solitary or multiple), surgeon's estimate of survival, presence/absence of visceral metastases and lymph node metastases, preoperative hemoglobin (on admission), absolute lymphocyte count and the primary oncologic diagnosis according to the method described by Katagiri et al. [32] with slight modifications. A Bayesian model is a statistical method used to explore the conditional, probabilistic relationships between variables to estimate the likelihood of an outcome using observed data. It can effectively account for uncertainty within the data and can be used in the setting of incomplete or missing input,

very much like the clinical scenario. Following its development, the model was found to be suitable for the clinical setting by performing a decision curve analysis [37]. In 2012, it was externally validated using Scandinavian registry data (*n* = 815) in patients with operable skeletal metastases of the extremities between 1999 and 2009 [28]. The models accurately predicted patient survival despite the varying amounts of missing data and different populations and made it publicly available to the international community at www.pathfx.org (accessed on 3 April 2022). Subsequently, it was revalidated using Italian (*n* = 287) [38] and Japanese (*n* = 261) [39] cohorts surgically treated for axial and appendicular MBD, retaining discriminatory ability and more importantly, clinical utility despite key differences in patient characteristics. In 2016, the models were stretched out to include 1-month and 6-month survival estimates using the training set [40] aiming to expand the risk stratification of patients in clinical and research settings, given the enlarging array of palliative treatment and mixed literature regarding 6- or 12-month patient survival as necessary life expectancies for endoprosthetic reconstruction. The same group validated PATHFx v2.0 using open-source machine learning software, which estimated the likelihood of 1-, 3-, 6-, 12-, 18- and 24-month survival and externally validated each model in two patient populations [41]. The original PATHFx models were created before immunotherapy, targeted therapies or checkpoint inhibitors became widely used, affecting primary tumor group allocation for a certain diagnosis, particularly in the case of NSCLC. Therefore, in 2019, Anderson et al. generated an updated PATHFx version 3.0 model using recent data (*n* = 208) combined with the original training set and including more prognostic features.

The model was externally validated using two contemporaries, but different patient populations, who were treated either surgically (*n* = 197) or non-surgically (*n* = 192) with external beam radiotherapy alone for symptomatic skeletal metastases. The study's results showed a favorable Brier score (>0.2), as well as DCA, indicating that it is better to use the models, rather than assume all or none of the patients with skeletal metastatic disease treated with surgical fixation will survive greater than 3, 6,12,18, 24- months. However, this was not the case in the 1-month model where it is better to assume patients will survive for 1 month than to rely solely on the PATHFx model. Additionally, AUC estimates were all greater than 0.70, which is suggestive of good discriminatory ability with lower bounds of the 95% CI all greater than 0.68, except for the 1-month radiotherapy-only group.

In order to extend the lifecycle of PATHFx through continued model improvement, it is currently linked to a large international registry, the International Bone Metastasis Registry [29].

### 3.2.7. IOR (Instituto Orthopedico Rizzoli) Score

In 2021, Errani et al [31] published the IOR score based on a prospective-based cohort (*n* = 159) study that analyzed patients with symptomatic MBD of the appendicular skeleton treated between 2015 and 2018 with a minor (INF etc.) or major surgery (endoprosthesis). The authors dichotomized the primary tumor profile proposed by Bollen et al. [35], considering survival at 12 months as a useful endpoint for surgical decision-making between endoprosthesis or internal fixation. In the study, unfavorable and moderate tumors reported a survival rate of 3.1 and 7.7 months, respectively; thus, both were categorized into the poor prognostic group. Univariate Kaplan Meier and multivariate Cox regression analyses of laboratory parameters and other possible prognostic features highlighted that pathological CRP ($\geq$1.0 mg/dL) and primary tumor diagnosis (bad prognosis) were the only significant negative prognostic variables equally associated with survival at 12 months after surgery. Patients were classified into three distinct prognostics based on the combination of the two later prognostic factors, each group with a different probability of survival. AUC estimates showed good performance of the model predicting survival at 12 months [31]. In 2021, Alfaro et al. validated the IOR score model in a multicenter retrospective study on a series of 136 patients with appendicular MBD who were treated surgically in a Chilean population. The study compared the performance of different survival prognostic models (revised Katagiri, PathFx, Optimodel and IOR score) and concluded that IOR score is the most accurate prognostic model for predicting a survival time of 12 months [17].

## 4. Discussion

Preoperative survival estimation is an important factor in the appropriate selection of patients who could possibly benefit from surgery for MBD of the extremity. Different prognostic models ranging from classic scoring models to machine learning algorithms have been developed as a support tool to predict mortality at different time points in patients who undergo surgical treatment of a bone metastasis of the extremity, of which some have been externally validated [23,25,26,28–31]. At present and in line with other areas in which prediction models are developed and applied, validation is expressed in metrics of discrimination and calibration [42].

Our review included seven of the most recent, different prognostic models that predict the preoperative survival of patients with appendicular MBD. According to current available studies, among externally validated survival prediction scores, the SPRING 2013 nomogram and PATHfx 3.0 model had a better performance status in terms of discrimination and accuracy. SPRING 2013 performed slightly better at predicting survival at 6 and 12 months in comparison to PATHfx 3.0, but performed similarly at the 3-month endpoint. It is interesting to note that they are both updated versions of previously created prognostic models. However, Mearns et al. showed superiority to the OptiModel in terms of performance at predicting survival at 12 months over SPRING 2013 using the same external validation database. The latter study showed PATHfx 1.0 to be more accurate at 3- and 6-month time periods.

Other more qualitative considerations of prognostic models should be noted.

Many of the studies and external validation cohort analyses were extracted in a retrospective design. With this design, uniformity in diagnostics and operative treatment criteria is not possible. Differences in local treatments between centers and over time are possible. IOR was the only study with a prospective design, but it was based on a relatively small cohort, which is another limitation of most of the reviewed models, making it hard to apply their conclusion in a more generalized way. Mearns et al. noted, by comparing six prognostic models using the same external validation database, differences between the study database and the Janssen and SPRING development datasets, which, in turn, led to reduced performance. Another drawback of retrospective studies is missing data. Willeumier et al. [26] used an interpretation of the clinical description to overcome a massive lack of KPS in his database, exposing the model to bias and decreased accuracy. Thio et al. and Sorensen et al. [24,25,27]. dropped variables with missing data, compromising the discriminatory abilities of the models. The Bayesian network, a machine learning method used by Pathfx 1.0 and Pathfx 3.0 [28,29], performed well despite differing patient populations and varying amounts of incomplete or missing data in several external validations, highlighting its ability to effectively account for uncertainty within the data [37–40].

Unfortunately, there is no consensus regarding which variables should be included in survival estimation models. Primary tumor histology is incorporated in all included models as a prognostic factor that significantly impacts survival but there is considerable variability in its content and subdivision. Some studies subdivide it according to Bollen et al. [26,31] and some according to Katagiri et al. [27–29]. Many models include multiple myeloma as the primary tumor [23–25,28–30], while others, such as Willeumier et al. [26], do not, claiming primary haematological cancer have a very different effect on survival than osseous metastases from solid carcinomas. The different effect on survival between the impending and pathologic fractures, found by Bauer and Wedin [10], was also reported by some recent prognostic models [24,25,28,29]; other studies did not find it significant [23,26,30,31]. Some studies identified laboratory factors, such as haemoglobin level [27–29], absolute lymphocyte count [28–30], platelets count, ALP and albumin [27] or CRP [31], as important preoperative predictors of survival. As expected, there is a degree of collinearity between laboratory factors and other clinical variables that can make them interchangeable. Willeumier et al. used only three variables in their model, aiming to create a clinically convenient model in comparison to numerous variable-based models [25,28–30], accepting the trade-off of reduced discriminatory ability for the simplicity of this prognostic model. Machine learning-based models address

these issues using a kitchen sink approach (having the algorithm select numerous variables) to improve the accuracy on one hand, while providing an easy-to-use, accessible tool for the clinician on the other hand. This variability in prognostic factor consideration impairs the quality and reliability of comparative studies in different populations and should be clearly clarified in the future.

Our study excluded publications that only include cohorts before 2005, arguing that the applicability of the prognostic model depends on remaining relevant as more effective systemic treatments are introduced [26,28,29]. The clinical profile ensures the sustainability of the model because of its dynamic description; it encompasses not only tumor growth speed but also contributing factors, such as the effectiveness of evolving systemic treatments and the rise of different subtypes in the various primary tumor. All models included in our review used clinical profiles as a tool for that purpose. Anderson et al. and Sorensen et al. refitted their original models, creating new versions based on larger and modern databases, maintaining their prediction models accurate and up-to-date [24,25,28,29].

The cohort's information in each prognostic model is remarkably different. More distinctly, patients included in these studies received different treatments and had different characteristics. Therefore, the prediction model may apply mainly to populations where decisions to surgically treated patients are approached in a similar fashion to the original model and differences may lead to poor performance status. For that reason, in the models where performance metrics were stated as AUCs, external validation cohorts showed reduced accuracy compared to the internal validation for all survival endpoints [24,25,28,29,31], as demonstrated in the literature [43]. The Janssen score, OptiModel and SPRING 2013 nomogram were externally validated twice while Pathfx 1.0 has been externally validated in a number of different patient populations [16,17,37–39]. In our analysis, the SORG model showed the best performance status at the 3- and 12-month endpoints in their internal validation compared to other studies; however, this does not reflect the true performance and applicability in different populations since it was not externally validated.

Our study highlights the challenges of the recent surviving predictor models in symptomatic MBD patients of the extremity, in terms of performance and applicability in different cohorts. This reminds us that the final therapeutic decision cannot rely on risk prediction models alone, but should be made by interdisciplinary corporations of oncologists, radiologists and orthopedic surgeons. It should also encourage us to further investigate and compare the most salient models in future, prospective, multicenter studies using standardized treatment regimes.

## 5. Conclusions

The development of individualized, preoperative survival prediction models for patients with appendicular MBD is a powerful decision-making tool for orthopedic surgeons and usage of these models has increased over the last two decades; however, there has been limited improvement in terms of calibration accuracy and discriminatory power recorded. Currently, it is still a challenge to recommend any of the models as the standard for predicting survival in these patients, but based on currently available studies, it seems like PATHFx 3.0, 2013-SPRING and possibly Optimodel are the best models in terms of both discrimination accuracy and calibration scores. External validation in different populations should be performed before the widespread use of a prediction algorithm. At the moment, only the PATHfx model addresses the missing data issue, is clinically user-friendly and widely externally validated. We recommend that future, large, multicenter, prospective studies will compare the PATHfx 3.0, SPRING 2013 and OptiModel using the same external validation dataset.

**Author Contributions:** Writing—original draft preparation, O.B.G., T.C.F.S., S.G.; writing—review and editing, O.B.G., A.M., S.V., V.J. and S.G. All authors have read and agreed to the published version of the manuscript.

**Funding:** This research received no external funding.

**Conflicts of Interest:** The authors declare no conflict of interest.

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
