# Peer review of "The Prediction of Survival after Surgical Management of Bone Metastases of the Extremities—A Comparison of Prognostic Models"

_curroncol, doi:10.3390/curroncol29070373_

Round 1

Reviewer 1 Report

This is a really interesting study with high publishable impact.

I have very few comments since I enjoyed reading the paper and I fount it really interesting for the readers and future endeveour.

1) the inclusion criteria must be better described. 

2) the study design is not clear. It seems a meta analysis but with some bias

3) it completely missing the analytical part. I suggest to analysis the data with the random or fixed effect regarding only the studies who reported the sd and weighing the data for the sample size. 

Author Response

Thanks for the time and effort in reviewing our manuscript, it is most appreciated.

  1. The inclusion criteria section was edited, I hope it is more clear now.
  2. Our study aimed to conduct a summary of the recent data on the subject and highlight the leading models by comparing them, we didn't intend to do a formal meta-analysis. The Heterogeneity
    on the statistical method for performance evaluation,
    the various mode of scoring systems, the use of different
    time points and more, can all impose bias and discrepancy.
  3. Could you please elaborate, I did not fully understand the comment.

Reviewer 2 Report

The authors introduced about the current the development of individualized preoperative survival prediction models for patients with appendicular MBD. It is meaningful since there has been limited improvement in terms of calibration accuracy and discriminatory power recorded. In this manuscript, 3 different models were applied to published data for widely externally validation. It is concluded that only the PATHfx model addressed the missing data issue. The review include comprehensive discussion about current research, I suggest to accept the manuscript in present form.

Minor revision:

The authors should add the citation to the study models and ids in the tables.

Author Response

Thanks for the time and effort in reviewing our manuscript.

The citation was added as recommended.

Kind regards,

Ofir

Reviewer 3 Report

This is a review paper on an interesting topic.

The authors stated that they limited their analysis to cases of the extremities, but I believe that bone metastases are relatively rare in extremities alone and are often accompanied by metastases in axial bones in addition. It is necessary to clearly indicate whether the analysis in this paper omits such cases or includes them.

It is questionable whether it was necessary to limit the analysis to cases with extremity metastasis in the first place. The usefulness of this paper in clinical practice may be limited.

Is the ERP in Table 1 a mistake for EPR? Also, Table 1 uses abbreviations and bracketed numbers, which I think must be annotated.

Author Response

Thanks for the time and effort in reviewing our manuscript.

Our review and more recent studies deal with symptomatic  MBD in the extremity separately due to differences in morbidity such as neurologic deficit and treatment options that can all influence the performance of each model. Since we aimed to compare prediction models we tried to minimise these differences knowing it can effect the generalisability of our conclusions.

EPR is annotated and the mistake in the table was revised.

Thanks again,

Kind regards,

Ofir

Reviewer 4 Report

To editors and reviewers
The prediction of survival after surgical management of bone metastases of the extremities - a comparison of prognostic models
- This is a very interesting manuscript that can be considered for publication in CURRENT ONCOLOGY. The manuscript is appropriate with aims and scope of journal. 
- I suggested some revisions below and after revisions the manuscript can be published.
1) Some citation and references are not precise as MDPI format. Please check and revise.

2) Abstract for a review should be unstructured.

3) Table 1 should add the year of publication in first column.

4) Figure 2 is quite small. Please enhance.

5) What is further direction research after this review. Please add one paragraph at the end of discussion.

6) The conclusion should be shown only in one paragraph. Please revise.

7) Other content is quite good without reviewer's concerns.

Sincerely

Author Response

Thanks a lot for the time to review our manuscript. It is most appreciated.

I have read your comments and suggestions and revised the manuscript as recommended :

  1. Done
  2. Abstract structure removed
  3. Years of publication added in table 1
  4. Figure 2 enhanced
  5.  A further direction research paragraph is added to discussion
  6. Conclusion revised 

Thanks,

Kind regards,

Ofir
